# Deep Reinforcement Learning for Sequential Combinatorial Auctions

## Abstract

Revenue-optimal auction design is a challenging problem with significant theoretical and practical implications. Sequential auction mechanisms, known for their simplicity and strong strategyproofness guarantees, are often limited by theoretical results that are largely existential, except for certain restrictive settings. Although traditional reinforcement learning methods such as Proximal Policy Optimization (PPO) and Soft Actor-Critic (SAC) are applicable in this domain, they struggle with computational demands and convergence issues when dealing with large and continuous action spaces. In light of this and recognizing that we can model transitions differentiable for our settings, we propose using a new reinforcement learning framework tailored for sequential combinatorial auctions that leverages first-order gradients. Our extensive evaluations show that our approach achieves significant improvement in revenue over both analytical baselines and standard reinforcement learning algorithms. Furthermore, we scale our approach to scenarios involving up to 50 agents and 50 items, demonstrating its applicability in complex, real-world auction settings. As such, this work advances the computational tools available for auction design and contributes to bridging the gap between theoretical results and practical implementations in sequential auction design.

## 1 Introduction

The effective allocation of scarce resources is a pervasive challenge across diverse domains, spanning spectrum licensing (FCC), transportation infrastructure (Rassenti et al., 1982), online advertising (Varian & Harris, 2014), and resource management (Tan et al., 2020). Combinatorial auctions (CAs) are a pivotal tool in addressing this challenge, offering a specialized auction format where bidders express valuations for combinations of items (or bundles). This allows for the incorporation of interdependencies among items, ultimately leading to more efficient allocations. For example, in spectrum auctions, bidders articulate preferences for combinations of licenses, capturing synergies and complementarities. However, despite their potential, CAs are known for their significant complexity, including computationally intensive winner determination problems and susceptibility to strategic bidding behavior (de Vries & Vohra, 2003).

*Sequential Combinatorial Auctions* (SCAs) make use of a sequential interaction with bidders — participants enter the auction in a predetermined order, strategically placing bids on available bundles that align with their interests. The sequential nature of SCAs yields several advantages. Primarily, it alleviates complexity by breaking down the problem into smaller, more manageable subproblems. Additionally, SCAs can be implemented as straightforward mechanisms with obvious strategyproofness guarantees (Li, 2017). The predetermined order of bidder arrivals plays a crucial role in ensuring no incentive for misreporting preferences, particularly when each stage's auction mechanism is designed to be strategyproof. Maintaining strategyproofness at each stage involves presenting bidders with a menu of options, allowing them to select their utility-maximizing choice. Such menu-based mechanisms enhance transparency and interpretability, simplifying the decision complexity for bidders who may not be experts in mechanism design.

Beyond the theoretical and practical advantages mentioned earlier, SCAs exhibit a surprising robustness in terms of generality. Cai & Zhao (2017) has demonstrated that there exists a straightforward SCA capable of achieving a constant approximation to optimal auctions when bidders' valuations

are XOS and $O(\log m)$-approximation in sub-additive setting. Despite this promising result, finding an optimal SCA mechanism remains an intricate task, primarily due to the vastness of its search space. Existing results, such as those by Cai et al. (2022), primarily focus on constructing a simple mechanism that achieves constant-approximation in a restricted XOS setting. In contrast, our research presents a general method for approximately identifying near-optimal mechanisms within a broader subadditive valuation settings.

Brero et al. (2020) introduced the use of reinforcement learning in auctions for the sequential setting, framing the design problem as a Markov Decision Process (MDP)—for example with the history of decisions so far constituting the *state*, and posted prices and which agent to visit next constituting the *action*. The reward is the payment collected from the bidder at each step. The overarching objective is to learn an optimal policy function that maps states to actions (the "mechanism,") maximizing the expected reward. To learn this policy function, Brero et al. (2020) use the Proximal Policy Optimization (PPO) algorithm. However, their focus is on a class of simple mechanisms known as *Sequential Pricing Mechanisms* (SPMs) with a straightforward menu structure where each item is associated with a posted price and the cost of a bundle is simply the sum of prices of its constituent items. In the realm of optimal multi-item auctions, bundling (beyond item-wise posted price) is a well-established necessity, even for simpler additive valuations. This calls for approaches that center on learning a more expressive menu structure that is capable of accommodating complex valuations. Consequently, in the MDP formulation, the action space needs to expand, posing challenges for directly applying standard RL methodologies to our setting.

**Main Challenges.** Most conventional reinforcement learning techniques are ill-suited for the large and continuous action spaces in this combinatorial auction setting. For instance, Q-learning based approaches are tailored for discrete action spaces. While one workaround involves discretizing the action space, this strategy proves impractical as it often involves evaluating all possible actions, which doesn't scale efficiently.

Although Proximal Policy Optimization (PPO) (Schulman et al., 2017), Soft Actor-Critic (SAC) (Haarnoja et al., 2018), and Deep Deterministic Policy Gradient (DDPG) (Lillicrap et al., 2016) can handle continuous action spaces, they often struggle with sample efficiency and convergence issues due to the curse of dimensionality that arise in extremely large action spaces. Policy gradient methods such as PPO suffer from high variance in gradient estimates, leading to convergence issues whereas Actor-Critic methods such as SAC, which involve learning a Q-function over both state and action spaces, require a prohibitively large number of samples to adequately cover their domain and generalize well. For these approaches to work, we would need extensive parallel environments (for samples) or extended algorithm runtime (for convergence), both of which are constrained by either hardware limitations or time constraints.

However, unlike traditional RL problems, the transition dynamics in sequential auctions can be accurately modeled, enabling the use of analytical gradients for more precise parameter updates. Such approaches involving analytical policy gradients (APG) have demonstrated high efficiency in domains such as differentiable physics and robotics (de Avila Belbute-Peres et al., 2018; Innes et al., 2019; Hu et al., 2019; Qiao et al., 2020; Wiedemann et al., 2023). However, existing APG methods usually require differentiable environments and rely on end-to-end training techniques such as backpropagation through time (BPTT), with gradients given by the simulator itself. This is not feasible in our framework, as the gradients are not readily available. Moreover, BPTT is prone to optimization challenges over long trajectories, such as exploding or vanishing gradients, further complicating its applicability.

Given these limitations, a new approach is needed to handle the challenges of continuous action spaces efficiently in combinatorial auctions. Our goal is to design an iterative method that leverages analytical gradients to overcome sample inefficiency, high variance and the computational complexities of existing methods.

**Our Contributions.** We introduce an new approach that uses *fitted policy iteration* (Bertsekas & Tsitsiklis, 1995; Bertsekas, 2011) and analytical gradients for learning revenue-maximizing sequential combinatorial auctions. This method uses neural networks to approximate the value function and policy function and iteratively update them in a twofold manner: initially refining the value function to align with the policy function and subsequently adjusting the policy function to maximize

rewards. While continuous action spaces pose challenges for the policy improvement step, we show, through Proposition 2, how we can simplify this to learning revenue-optimal single-buyer auctions via gradient descent.

We implement the policy improvement step by extending the neural network architectures for static revenue optimal auction design, such as RochetNet (Dütting et al., 2023) or MenuNet (Shen et al., 2019), to the sequential setting by modifying the menu structure to include an additional term called the "offset". The offset captures the value of potential future states ("continuation value"). By incorporating the offset into the optimization objective, our approach aims to maximize both the current revenue and the anticipated revenue from future states. This adapted network, trained through first-order gradient methods, offers a more effective and stable approach to policy iteration in continuous action spaces. Instead of parameterizing and learning menu options for each state and bidder, we learn the weights of a neural network that takes in as input a state and outputs the corresponding menu options. This let's us handle combinatorial auctions with up to 20 agents and a menu size of up to 1024.

Furthermore, we demonstrate the scalability of our approach to accommodate a large number of buyers and items, extending to as many as 50 buyers and 50 items for the additive-valuation setting, significantly surpassing the capabilities of existing methods based on differentiable economics for auction design. This scalability is achieved by learning the menus corresponding to the *entry-fee mechanisms*. While this is less expressive than a combinatorial menu structure, it is more computationally efficient, making it a viable and efficient solution for scenarios involving a larger number of buyers and items.

**Related Works.** Our work is closely related with the literature of sequential combinatorial auctions design (Cai & Zhao, 2017; Cai et al., 2022), in which the previous papers focus on the theoretical characterization of the approximation results. Cai & Zhao (2017) first proposed a simple sequential posted price with entry fee mechanism (SPEM) and proved that the existence of such mechanism achieves constant approximation to the optimal mechanism in XOS valuation setting and $O(\log(m))$ approximation in the subadditive valuation setting. Later work by Cai et al. (2022) provides a polynomial algorithm based on linear program (LP) to compute the simple mechanism to achieve constant approximation in the item-independent XOS valuation setting. Compared to these existing literature, our work focuses on finding near-optimal SCAs with a general auction format (e.g., we allow bundle pricing rather than item pricing) through the use of deep learning based approaches. In addition, there is a rich literature designing approximation-results for online combinatorial auctions using simple posted-price mechanism (through Prophet Inequality), in which the items arrive in a sequential manner (Feldman et al., 2015; Dütting et al., 2020; Assadi & Singla, 2019; Deng et al., 2021). Another loosely related research direction is dynamic mechanism design (Ashlagi et al., 2016; Papadimitriou et al., 2022; Mirrokni et al., 2020), where the previous papers focus on mechanism design problem dealing with forward-looking agents that the agents may deviate their truthful reporting at current rounds to get more utility in the long-run.

The application of deep learning to auction design has garnered significant research attention in recent years, opening up exciting new avenues for achieving optimal outcomes. The pioneering work by Dütting et al. (2023) demonstrated the potential of deep neural networks for designing optimal auctions, recovering known theoretical solutions and generating innovative mechanisms for the multi-bidder scenarios. Subsequent research extended the original neural network architectures, RegretNet and RochetNet proposed in Dütting et al. (2023), to specialized architecture to handle IC constraints (Shen et al., 2019), to handle different constraints and objectives (Feng et al., 2018; Golowich et al., 2018), adapt to contextual auctions setting Duan et al. (2022), and incorporate with human preference (Peri et al., 2021). In addition, there are other research efforts to advance the training loss of RegretNet (Rahme et al., 2021b), explore new model architectures(Rahme et al., 2021a; Duan et al., 2022; Ivanov et al., 2022), certify the strategyproofness of the RegretNet Curry et al. (2020) and extend RochetNet framework for Affine Maximizer Mechanisms for the setting that there are dynamic number of agents and items (Duan et al., 2023). Recently, Zhang et al. (2023) apply deep learning techniques to compute optimal equilibria and mechanisms via learning in zero-Sum extensive-form games, in the setting when the number of agents are dynamic.

There is also previous interest in applying deep reinforcement learning (DRL) to auction design. Shen et al. (2020) propose a DRL framework for sponsored search auctions to optimize reserve

price by modeling the dynamic pricing problem as an MDP. DRL has also been used to compute near-optimal, sequential posted price auctions (Brero et al., 2020; 2023), where the authors model the bidding strategies of agents through an RL algorithm and analyze the Stackelberg equilibrium of the sequential mechanism (perhaps also allowing for an initial stage of communication). Meanwhile, Gemp et al. (2022) investigated the use of DRL for all-pay auctions through conducting the simulations from the multi-agent interactions. Existing papers using DRL for auction design focus on additive or unit- or additive-demand valuation settings. Whereas, in our paper, we propose a general, sequential combinatorial auction mechanism through DRL, which has potential to handle much larger combinatorial valuation settings by utilizing the sequential auction structure along with our customized DRL algorithm.

## 2 PRELIMINARIES

We consider a setting with $n$ bidders, denoted by the set $N = \{1, \ldots, n\}$ and $m$ items, $M = \{1, \ldots, m\}$. Each bidder $i \in N$ has a valuation $v_i : 2^M \to \mathbb{R}$, where $v_i(S)$ denotes the valuation of the subset $S \subseteq M$. Each bidder valuation $v_i$ is drawn independently from a distribution $\mathcal{V}_i$. We denote $\mathcal{V} = \Pi_{i=1}^n \mathcal{V}_i$.

We consider the sequential setting where the auctioneer visits the bidder in lexicographical order. He knows the distribution $\mathcal{V}$ but not the bidder's private valuations $v_i$ for $i \in N$. The mechanism design goal is to design a set of allocation rules and payment rules that determine how these items are allocated and how much each bidder is charged such that the expected revenue (gross payment collected from the bidders) is maximized. We denote the allocation rule by $g = (g_1, \ldots, g_n)$ where $g_i \subseteq M$, denotes the subset of items allocated to bidder $i$. Since the items can't be over-allocated, we require $g_i \cap g_j = \varnothing$ for all $i \neq j \in N$. We denote the payment rule as $p = (p_1, \ldots, p_n)$ where $p_i \in \mathbb{R}_{\geq 0}$ denotes the payment collected from bidder $i$.

In this work, we study Sequential Combinatorial Mechanisms with Menus. Given a bidder $i$ and a set of available items denoted by $S$, the mechanism consists of a pricing function $\xi_{i,S} : 2^M \to \mathbb{R}_{\geq 0}$ for each bidder $i$ and the set of available items $S$ that maps a subset of items (i.e. bundle) to its price. Additionally, we require $\xi_{i,S}(T) = \infty$ if $T \nsubseteq S$ to prevent the over-allocation of items. The auctioneer engages with bidders in lexicographic order, presenting them with a menu of bundles along with their corresponding prices. Subsequently, the bidder being visited selects their preferred bundle, pays the associated price, and exits the auction. The favorite bundle for the bidder is simply defined as the bundle that maximizes their expected utility[1] i.e. $S_i^* = \arg\max_{T \subseteq S} v_i(T) - \xi_{i,S}(T)$. See Algorithm 1 for the details of the mechanism.

---
**Algorithm 1** Sequential Combinatorial Mechanisms with Menus

---
**Require:** $\xi_{i,S}(\cdot)$ is bidder $i$'s pricing function when the menu consists of bundles of items from set $S$
1: $S \leftarrow [m]$
2: **for** $i \in [n]$ **do**
3:     Show bidder $i$ the menu $\mathcal{M}$ of available bundles and their corresponding prices, i.e. $\mathcal{M} = \{(T, \xi_{i,S}(T)) \mid T \subseteq S\}$
4:     $i$ picks their favorite bundle $S_i^* \subseteq S$, and pays $\xi_{i,S}(S_i^*)$.
5:     $S \leftarrow S \backslash S_i^*$.
6: **end for**

---

**Remark 1.** The mechanism described in Algorithm 1 is Dominant Strategy Incentive Compatible (IC) and Individual Rational (IR) if $\xi_{i,S}(\varnothing) = 0$ for all $i \in [n], S \subseteq M$.

This menu structure is more expressive than the Sequential Posted Price with Entry Fee Mechanism (SPEM) (Cai & Zhao, 2017) and Sequential Price Mechanisms (SPM) (Brero et al., 2020). In a SPEM mechanism, every bidder $i$ at a particular time step is shown the set of available bundles $S$ and the posted prices $p_{i,j}$ for every item $j$ in $S$ and is charged an entry-fee $\delta_i(S)$ to participate. If the bidder accepts, she can picks any bundle $T \subseteq S$ by paying an additional $\sum_{j \in T} p_{i,j}$. In SPMs,

---
[1]Ties are broken in favor of a bigger subset

bidder $i$ is shown a set of available items $S$ and is charged $\sum_{j \in T} p_{i,j}(S)$. The price here depends on the state; however, there is no entry fee.

Given the sequential nature of this problem, we follow Brero et al. (2020) and formulate learning the pricing function as a finite horizon Markov Decision Process (MDP). This MDP is defined by:

- **State Space** $\mathcal{S}$: The state $s^t$ at each time step $t$ is a tuple consisting of bidder under consideration and the set of items remaining at time $t$. We have $s^t = (i^t, S^t)$.

- **Action Space** $\mathcal{A}$: The action $a^t$ at each time step $t$ is a vector of bundle prices. We thus have $a^t \in \mathbb{R}^{2^M}$ where $a_T^t$ denotes the price associated with bundle $T$. We have $a_T^t = \xi_{s^t}(T)$.

- **State Transitions:** The agent $i^t$ under consideration chooses their favorite bundle based on the realized private valuation $v_{i^t}$, and state becomes $s^{t+1} = (i^{t+1}, S^{t+1})$ where $S^{t+1} = S^t \setminus \arg\max_{T \subseteq S^t} v_{i^t}(T) - a_T^t$. The stochasticity is in the private valuation $v_{i^t} \in \mathcal{V}_{i^t}$.

- **Reward** $r : \mathcal{S} \times \mathcal{A} \times \mathcal{S} \to \mathbb{R}$: The reward is simply the payment collected from the bidder at time $t$. This is given by price associated with the bundle picked by agent $i^t$. We have $r(s^t, a^t, s^{t+1}) = a_{S^t \setminus S^{t+1}}^t = \xi_{s^t}(S^t \setminus S^{t+1})$.

The discount factor $\gamma = 1$. Let $\pi : \mathcal{S} \to \mathcal{A}$ denote the policy function that maps the states to actions. Let $\rho_\pi(s^t, a^t, s^{t+1})$ denote the state-action-state marginals of the trajectory induced by $\pi(s^t)$. The objective here is to learn a policy function $\pi_*$ that maximize expected rewards i.e $\pi_* = \arg\max_\pi \sum_{t=1}^n \mathbb{E}_{(s^t, a^t, s^{t+1}) \sim \rho_\pi} \left[ r(s^t, a^t, s^{t+1}) \right]$. Let $V_\pi : \mathcal{S} \to \mathbb{R}$ denote the value function. $V_\pi(s^t) = \sum_{t'=t}^n \mathbb{E}_{(s^{t'}, a^{t'}, s^{t'+1}) \sim \rho_\pi} \left[ r(s^{t'}, a^{t'}, s^{t'+1}) \right]$ i.e. $V_\pi(s^t)$ is the expected sum of rewards if we start at state $s^t$ and act according to policy $\pi$. Let $V_*$ denote the optimal value function i.e. $V_* = V_{\pi^*}$. For the sake of notational convenience, we define $V_\pi(s^{n+1}) = 0$ for all policy functions $\pi$. Additionally, the policy function is, in essence, the bundle pricing function. Thus, we have $\pi_T(s^t) = \xi_{s^t}(T)$, where by $\pi_T(s^t)$ we mean the coordinate of $T$ in the vector $a^t = \pi(s^t)$.

Given this MDP, one can use tools (such as Gymnasium (Towers et al., 2023)) to write an environment that simulates the behavior of the bidders and an off-the-shelf reinforcement learning algorithm suitable for continuous action spaces (such as PPO or SAC) to learn the parameters of policy functions to maximize expected rewards. As discussed in Section 1, these traditional RL approaches encounter several issues in a large action space setting and our empirical findings corroborate the same. It is important to highlight that in our specific context the action space is exceptionally high-dimensional, even for a reasonably small number of items, as it grows exponentially. For instance, even with just 10 items, the action space expands to a size of $2^{10}$.

Note that in our setting, given agents' realized valuation, we can accurately predict subsequent states. The randomness in this context arises from these realized values, but we know their distributions a priori. This allows us to model the transition functions differentiably. Leveraging this unique advantage, we design new approaches that use first-order gradient methods along with policy iteration to learn menus for different states.

## 3 METHOD

Policy iteration consists of two alternating steps — a *policy evaluation* step that computes a value function consistent with the current policy, and a *policy improvement* step that uses the value functions to greedily find better policies. Each iteration is a monotonic improvement over the previously computed policies and often converges quickly in practice. For finite-state MDPs with a finite action space, it has also been shown to converge to the optimal policy. However, we are dealing with continuous action spaces for which the policy improvement step is intractable. In this section, we show how we address this challenge.

Since we know the transition dynamics, we can use it to simplify this step. Given a state $s^t$ and an action $a$, we can exactly estimate the next state conditioned on the realized private value $v \sim \mathcal{V}_{i^t}$. We now show how we can efficiently compute the best policies.

**Proposition 2.** *For a current policy $\pi$ and value function $V_\pi(.)$, the improved policy $\pi'$ for a state $s^t$ is given by:*

$$\pi'(s^t) = \arg\max_{a^t} \mathbb{E}_{v \sim \mathcal{V}_{i^t}} \left[ a^t_{S^t_*(v)} + V_\pi(i^{t+1}, S^t \setminus S^t_*(v)) \right] \tag{1}$$

See Appendix B for the proof. When $t = n$, we have $V_\pi(i^{n+1}, S^n \setminus S^n_*(v)) = 0$ and the objective reduces to maximizing $\mathbb{E}_{v \sim \mathcal{V}_{i^n}} \left[ a^n_{S^n_*(v)} \right] = \mathbb{E}_{v \sim \mathcal{V}_{i^n}} \left[ \xi_{s^n}(S^n_*(v)) \right]$. This objective corresponds to learning a revenue maximizing auctions with a deterministic menu of options over items in $S^n$ i.e. $\mathcal{M} = \left\{ (T, \xi_{i^n, S^n}(T)) \mid T \subseteq S^n \right\}$ for a single buyer $i^n$. Note that this objective is not smooth due to the computation of $S^n_*(v)$. Following the training of RochetNet, we relax our objective to a smoother version: $\mathbb{E}_{v \sim \mathcal{V}_{i^n}} \left[ \sum_{T \in S^n} \Delta_T(v) \cdot a^n_T \right]$. Here, $\Delta(v)$ is a softmax over the utilities of picking different subsets for a given valuation $v$. We optimize this objective using samples drawn from agent $i^n$'s value distribution over items in $S^n$. While computing this softmax, we scale the utilities by $\frac{1}{\tau}$ where $\tau$, a hyperparameter, controls the degree of smoothness. As $\tau \to 0$, $\Delta(v)$ converges to a one-hot vector with a value $1$ at the index corresponding to the utility-maximizing bundle thus recovering our original objective.

For $t < n$, we need to account for the additional non-zero term while solving the maximization problem. Every menu option now consists of a bundle, its corresponding price, and the value of the next state if the bundle is picked. While there is no change to how the utility-maximizing bundles are picked, this "offset" term corresponding to the picked bundle is taken into account while maximizing revenue. The smoother objective for this case is thus $\mathbb{E}_{v \sim \mathcal{V}_{i^n}} \left[ \sum_{T \in S^n} \Delta_T(v) \cdot (a^t_T + \phi(T)) \right]$. Here, $\phi(T) = V_\pi(i^{t+1}, S^t \setminus T)$ represents the offset, which is the expected revenue from future states if bundle $T$ is picked by $i^t$.

We show how we can implement the policy improvement step in Algorithm 2.

---

**Algorithm 2** Policy Improvement Step

---

**Require:** Hyperparameters $\tau, \Gamma, \eta, \ell$
1: **function** PI$\left( \left( s_t, V_\pi(i^{t+1}, \cdot), \mathcal{V}_{i^t} \right) \right)$
2: $\quad a_T \leftarrow 0, \forall T \subseteq S$
3: $\quad \phi(T) \leftarrow V_\pi(i^{t+1}, S^t \setminus T), \forall T \subseteq S$
4: $\quad$ **for** iter $\in 1, \dots, \Gamma$ **do**
5: $\quad\quad$ Receive $\{v_1, \dots, v_\ell\} \sim \mathcal{V}_{i^t}$
6: $\quad\quad$ Construct menu $\mathcal{M} = \{T, a_T, \phi(T)\}_{T \subseteq S}$
7: $\quad\quad u_{j,T} \leftarrow v_j(T) - a_T \, \forall T \subseteq S^t, \forall j \in [\ell]$
8: $\quad\quad \Delta_{j,\cdot} \leftarrow \mathsf{softmax}(u_{j,\cdot}/\tau)$
9: $\quad\quad \mathcal{L}(a) \leftarrow -\sum_{j \in \ell} \sum_{T \subseteq S^t} \Delta_{j,T} \cdot (a_T + \phi(T))$
10: $\quad\quad a \leftarrow a - \eta \nabla \mathcal{L}(a) \, \forall T \subseteq S^t$
11: $\quad$ **end for**
12: $\quad \pi_T(s^t) \leftarrow a_T \, \forall T \subseteq S^T$
13: $\quad V_\pi(s^t) \leftarrow \sum_{j \in \ell} (a_{S^t_*(v_j)} + \phi(S^t_*(v_j)))$
14: $\quad$ **Return** $\pi_T(s^t), V_\pi(s^t)$
15: **end function**

---

Given a policy $\pi$, it's straightforward to estimate the value function Monte-Carlo and Temporal Difference Learning methods. Since we know the transition dynamics, we can use this to further improve our value function estimates. For a state $s_t$, the value function can be estimated as follows:

$$V_\pi(s_t) = \mathbb{E}_{v \sim \mathcal{V}_{i^t}} \left[ \pi_{S^t_*(v)}(s_t) + V_\pi(i^{t+1}, S^t \setminus S^t_*(v)) \right] \tag{2}$$

## 3.1 EXACT METHOD FOR SMALL NUMBER OF STATES

When the number of states is small, we can use dynamic programming to solve for the optimal policy. Since the states are never repeated for a given episode and policy improvement steps for a state $s^t$ only depend on future time steps, we can start by computing the policy improvement for all possible states at $t = n$ and proceed backward to $t = 0$. To be more precise, for a state $s^t = (i^t, T)$ for $T \subseteq [m], t = n$, we perform the policy improvement step to compute $\pi(s^t)$. We set $V(s^t)$ to

be the revenue computed from this step. Next, we repeat this process for all states $s^t = (i^t, T)$ such that $T \subseteq [m], t = n - 1$ and proceed until we get to the state $(i^1, [m])$. In total, this involves training $n \times 2^m$ RochetNets, one for each state. For more details, refer to Algorithm 3

---

**Algorithm 3** Dynamic Program (DP)

---
**Require:** $m, n, \mathcal{V}$
1: $S \leftarrow [m]$
2: $V_\pi(i^{n+1}, T) \leftarrow 0 \; \forall T \subseteq S$
3: **for** $t \in n, n-1 \ldots, 1$ **do**
4:      **for** $s_t \in S$ **do**
5:          $\pi_T(s^t), V_\pi(s^t) \leftarrow \text{PI}(s^t, V_\pi(i^{t+1}, \cdot), \mathcal{V}_{i^t})$
6:      **end for**
7: **end for**

---

### 3.2 Approximate Methods for Large Number of States

When the number of states is large, we use a feed-forward neural network (called the critic) to map a state to its value. The critic network is denoted by $V^\alpha : N \times \{0,1\}^m \to \mathbb{R}$. We use another feed-forward network called the actor that maps a state to action. The actor network is denoted by $\pi^\theta : N \times \{0,1\}^m \to \mathbb{R}_{\geq 0}^{2^M}$.

**Neural Network Architecture.** The state $s^t$ forms the input to the critic as well as the actor network and is represented as an $m + 1$ dimensional vector. The first index denotes the index of the agent visited at time $t$ and the last $m$ indices are binary numbers that denote the availability of the corresponding item. The first index is used to compute a positional embedding of dimension $d_{emb}$ which is then concatenated with the $m$-dimensional binary vector to form a $m + d_{emb}$ dimensional input to the feed-forward layers. We use $R$ hidden layers with $K$ hidden units and a non-linear activation function for the actor as well as the critic. Note that these networks do not share any parameters.

The critic outputs a single variable capturing the value of the input state. The actor first outputs $p \in \mathbb{R}_{\geq 0}^{2^m}$ variables that correspond to the bundle prices. We use the soft plus activation function to ensure these outputs are positive. We use the $m$-dimensional binary input to compute a boolean mask variable $\beta \in \{0,1\}^{2^m}$ to denote the availability of a bundle. The final output of the actor network is given as $p \cdot \beta + (\vec{1} - \beta) \cdot K$ where $K$ is a large constant and the product denotes entry-wise masking. This masking ensures that unavailable bundles are assigned a high price ensuring that they are never picked, thereby satisfying feasibility constraints.

**Fitted Policy Iteration.** To perform policy iteration, we first randomly initialize the critic network (value network). Then we perform the approximate policy evaluation step. To do this, we first collect several state-action-reward samples and store them in a buffer. To encourage exploration, we add Gaussian noise to the action while collecting the samples. We reduce the magnitude of added noise by a small amount for later iterations. We use TD($\lambda$) to compute expected returns $V_{targ}^t$ for every state $s^t$ in the state-action-reward samples. We use these as targets and minimize an MSE loss to fit the critic to the computed values. We also found it helpful to update $V_{targ}^t$ using Equation 2 after a few initial critic network iterations, and then continue updating the critic network with the new targets.

Once we have trained the critic, we perform the approximate policy improvement step. To do so, we update the weights of the actor (policy network) to minimize the expected actor loss over all states in the buffer. The actor loss, denoted by $\mathcal{L}_\pi^\theta$, for a state $s^t$ is simply the negative expected revenue that the policy achieves from state $s^t$. Thus we have:

$$\mathcal{L}_\pi^\theta(s^t) = -\sum_{j \in \ell} \sum_{T \subseteq S^t} \Delta_{j,T} \left( \pi_T^\theta(s^t) + V^\alpha(i^{t+1}, S^t \setminus T) \right) \tag{3}$$

where $\Delta_{j,T}$ is defined in Line 8 of Algorithm 2. Refer to Figure 1 for an overview of the loss computation. We repeat the policy evaluation step and policy iteration step until convergence. For more details, refer to Algorithm 4.

**Algorithm 4** Fitted Policy Iteration (FPI)

**Require:** $\eta_n, \eta_\pi, \eta_\epsilon, \epsilon_0$
1: Initialize neural network parameters $\theta, \alpha$
2: Initialize noise $\epsilon \leftarrow \epsilon_0$
3: Initialize rollout buffer $\mathcal{D}$
4: **for** each iteration **do**
5:     **for** each `environment_step` **do**
6:         $noise \sim \mathcal{N}(0, \epsilon)$
7:         $a^t \leftarrow \pi^\theta(s^t) + noise$
8:         $s^{t+1} \sim p(s^{t+1}|s^t, a^t)$
9:         $\mathcal{D} \leftarrow \mathcal{D} \cup \{s^t, a^t, r(s^t, a^t, s^{t+1}), s^{t+1}\}$
10:    **end for**
11:    Compute $V_{targ}^t$ through TD-$\lambda$
12:    **for** each `critic_gradient_step` **do**
13:        $J_v(\alpha) = \sum_{s^t \in \mathcal{D}} \|V_{targ}^t - V^\alpha(s^t)\|^2$
14:        $\alpha \leftarrow \alpha - \eta_v \nabla J_v(\alpha)$
15:    **end for**
16:    **for** each `actor_gradient_step` **do**
17:        $J_\pi(\theta) = \sum_{s^t \in \mathcal{D}} \left[ \mathcal{L}_\pi^\theta(s^t) \right]$
18:        $\theta \leftarrow \theta - \eta_\pi \nabla J_\pi(\theta)$
19:    **end for**
20:    $\epsilon \leftarrow \epsilon \eta_\epsilon$
21: **end for**

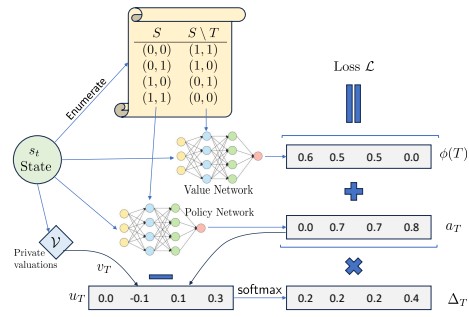

**Figure 1:** Loss function for the policy improvement step. Given a state, we first enumerate the menu options and the corresponding future states. We use the actor to compute the prices and the critic to compute the offsets. With this we compute the loss in Equation 3

### 3.3 ENTRY FEE MECHANISMS FOR EXTREMELY LARGE NUMBER OF STATES

Note that the current menu structure that we impose involves learning $2^m$ bundle prices per state. The computation of which bundle is utility maximizing requires $O(2^m)$ computations. To scale up our approach to a large number of items, say $m > 10$, we show how we can impose the menu structure of entry-fee mechanisms. This involves learning $m + 1$ values — posted prices $p_i$ for $i \in [m]$ for the items and an entry fee $\delta$. The price of bundle $T$ is given by $\delta + \sum_{i \in T} p_i$. When the agent valuations are additive, computing the utility-maximizing bundle only requires $O(m \log m)$ computations.

To see this, consider an additive valuation represented by $t = (t_1, \ldots t_m)$. The value for any bundle $T$, denoted by $V(T)$, is given by $\sum_{i \in T} t_i$. To compute the utility-maximizing bundle, sort the vector given by $(t_1 - p_1, \ldots, t_m - p_m)$ to compute indices $\sigma_1, \ldots, \sigma_m$. We thus have $t_{\sigma_1} - p_{\sigma_1} \geq \ldots t_{\sigma_m} - p_{\sigma_m}$. Construct a menu of $\mathcal{M} = \{T_k, p_k\}_{k \in [m]} \cup \{\varnothing, 0\}$ where $T_k = \{\sigma_1, \ldots, \sigma_k\}$ and $p_k = \delta + \sum_{i \in k} p_{\sigma_i}$. The utility-maximizing bundle is given by $\max_{k \in [m+1]} V(T_k) - p_k$ (where for convenience we denote $T_{m+1} = \varnothing$ and $p_{m+1} = 0$). Instead of using masks to set unavailable bundle prices to a large value $K$, we simply set the posted price of an unavailable item to $K$ to satisfy allocation constraints. We only have to compute utility and the softmax values of these $m$ options (instead of $2^m$) while training the actor networks.

## 4 EXPERIMENTAL RESULTS

In this section, we present experimental results, comparing our approach with two established analytical mechanism design baselines: the *item-wise* and *bundle-wise* sequential auctions. The former sells items individually, while the latter treats all items as a unified bundle, selling it sequentially. The optimal policy for both these methods can be computed through Dynamic Programming. Additionally, we benchmark our approach against other reinforcement learning algorithms such as PPO. Approaches involving Q-function learning, like SAC and DDPG, were unstable and did not perform as well as PPO. Notably, we observe that the size of the action space significantly influences training, with a vast action space making it impractical to accurately evaluate and update Q-values for all possibilities. Across all our settings, we provide results based on 10,000 episodes. Additional details regarding implementation and hyperparameters can be found in the Appendix C. Next, we elaborate on the various settings considered in this paper.

| # Agents $n$ | # Items $m$ | Setting | Action Space Size | MD Baselines | | RL Baselines | | Ours | |
|---|---|---|---|---|---|---|---|---|---|
| | | | | Item-wise | Bundle-wise | PPO | SAC | FPI | DP |
| 5 | 5 | A | 32 | 3.00 | 2.58 | 3.09 | 2.74 | *3.12* | **3.13** |
| | | B | 32 | 1.80 | 1.56 | 1.83 | 1.60 | *1.86* | **1.87** |
| | | C | 6 | *2.42* | – | **2.43** | **2.43** | **2.43** | **2.43** |
| | | D | 26 | 3.00 | 1.83 | 3.08 | 2.80 | *3.10* | **3.11** |
| 10 | 10 | A | 1024 | 7.41 | 5.57 | 6.68 | 4.88 | *7.59* | **7.60** |
| | | B | 1024 | *4.07* | 3.11 | 3.80 | 2.64 | **4.16** | **4.16** |
| | | C | 11 | 5.92 | – | *5.96* | *5.96* | **6.00** | **6.00** |
| | | D | 176 | *7.37* | 6.06 | 7.11 | 6.70 | **7.54** | **7.54** |

**Table 1:** Test Revenue achieved by different approaches for the Constrained Additive Setting.

## 4.1 CONSTRAINED ADDITIVE VALUATIONS

Let the values of individual items be denoted by $t_j \sim \mathcal{V}_j$. In the *additive valuation* case, the value of a bundle is calculated as the sum of the individual values of its constituent items: $V(S) = \sum_{j \in S} t_j$. For *unit-demand valuation*, each bidder values only their most preferred item in a bundle, making the value of the bundle equal to the highest-valued item within it: $V(S) = \max_{j \in S} t_j$. In the context of *k-demand valuation*, each bidder values the $k$ most preferred items in a bundle, with the value of the bundle determined by the highest-valued item among these $k$: $V(S) = \max_{R \subseteq S, |R|=k} t_j$.

We consider the following settings each with $n$ agents and $m$ items with $(n, m) \in \{(5,5), (10,10)\}$:

A. Additive valuations where item values are independent draws from $U[0, 1]$

B. Additive valuations where item $i$'s values are independent draws over $U[0, \frac{i}{m}]$

C. Unit-demand valuations where item values are independent draws over $U[0, 1]$

D. $k-$demand valuations where item values are independent draws over $U[0, 1]$ and $k = 3$

We present the results in Table 1. PPO performs adequately in smaller settings involving 5 agents and 5 items. However, its performance degrades when the scale is increased to 10 agents and 10 items. In contrast, our proposed method based on Dynamic Programming (DP) and Fitted Policy Iteration (FPI) consistently outperforms established baselines. Importantly, FPI achieves these results with a computational time of less than 20 minutes on a single Tesla A100 GPU. In comparison, PPO required several hours of training under these conditions. We terminated training PPO after 20,000 iterations, which took approximately 5 hours. The Dynamic Programming (DP) approach took 6 hours for $n = m = 10$. The training curves (expected revenue vs updates) for PPO and FPI are in Appendix D.

## 4.2 COMBINATORIAL VALUATIONS

In the combinatorial setting, we consider $n \in \{10, 20\}$ agents and $m = 10$ items with the following bundle-wise valuations listed below:

E. Subset valuations are independent draws over $U[0, \sqrt{|S|}]$ for every subset S

F. Subset valuations are given by $\sum_{j \in T} t_j + c_T$ where $t_j \sim U[1, 2]$ and the complimentarity parameter $c_T \sim U[-|S|, |S|]$

The results are presented in Table 2. Our approach, specifically designed to navigate high-dimensional spaces more effectively, outperforms PPO in combinatorial settings as well. Notably, our method (FPI) required only 20 minutes of training on a Tesla A100 GPU while DP took between 5 and 10 hours, depending on the number of agents. PPO was trained for 20,000 iterations, which took approximately 12 hours.

## 4.3 SCALING UP

We scale our approach to an even larger number of buyers and items. For this setting, we use the entry-fee mechanisms to characterize the menus, enhancing computational efficiency and reducing

| # Agents $n$ | # Items $m$ | Setting | Action Space Size | MD Baselines | | RL Baselines | | Ours | |
|---|---|---|---|---|---|---|---|---|---|
| | | | | Item-wise | Bundle-wise | PPO | SAC | FPI | DP |
| 10 | 10 | E | 1024 | 6.20 | 2.35 | 4.19 | 2.43 | *7.00* | **7.01** |
| | | F | 1024 | 22.25 | 21.69 | 23.82 | 15.16 | **24.48** | 24.43 |
| 20 | 10 | E | 1024 | 8.08 | 2.68 | 4.31 | 2.17 | *8.17* | **8.19** |
| | | F | 1024 | 24.29 | 22.82 | 24.57 | 14.72 | **25.70** | *25.48* |

**Table 2:** Test Revenue achieved by different approaches for the Combinatorial Valuations Setting.

memory requirements, as we only manage $m + 1$ menu options instead of $2^m$ options at any given time. We consider Setting A with $(n, m) \in \{(20, 20), (50, 50)\}$

We present the results in Table 3. We observe that that with an increasing number of agents, *item-wise Myerson* emerges as a competitive baseline for the additive valuation setting, with policy iteration showing only a slight performance edge. Note that PPO demonstrates significantly improved performance in this setting due to the smaller action-space associated with the entry fee mechanism.

| # Agents $n$ | # Items $m$ | Setting | Action Space Size | MD Baselines | | RL Baselines | | Ours | |
|---|---|---|---|---|---|---|---|---|---|
| | | | | Item-wise | Bundle-wise | PPO | SAC | FPI | DP (Sym) |
| 20 | 20 | A | 21 | 16.42 | 11.38 | 16.23 | 12.19 | *17.11* | **17.14** |
| 50 | 50 | A | 51 | 46.47 | 28.20 | 43.64 | 32.13 | **46.71** | *46.54* |

**Table 3:** Test Revenue achieved by different approaches for the Large Scale Setting.

For these settings, we also report the performance of DP, which was trained using a fully expressive menu but with symmetry imposed. This reduces the number of RochetNets trained from $n \times 2^m$ to $nm$ with each RochetNet having $m$ menu options (one for every bundle size) instead of $2^m$. We were unable to train without the imposition of symmetry due to hardware and time limitations.

## 5 CONCLUSION

We have introduced a new methodological approach to the problem of learning simple and strategyproof mechanisms for sequential combinatorial auctions. We formulate this as a reinforcement learning problem and show how we can use first order gradients to learn menu options that maximize expected revenue. Through extensive experimental results, we've shown the superior performance of this approach compared to other RL methods and well-known analytical solutions.

We also point out some limitations and potential directions for future work. The use of neural networks in approximate methods introduces uncertainties, lacking theoretical guarantees for convergence. Additionally, the optimality of solutions obtained by RochetNet in the policy improvement step remains unknown. Despite these uncertainties, we empirically observe convergence in all our experiments, and it has been shown that RochetNet consistently retrieves optimal solutions when analytical solutions are available (Dütting et al., 2023). Another limitation lies in our dependence on the assumption of having ample samples from valuation distributions. While this is a common practice in empirical approaches to mechanism design, it would be insightful to explore the effectiveness of our approach when the number of available samples is limited.

Future work could also explore more intricate menu structures beyond entry-fee mechanisms, continuing to seek computational efficiency improvements over bundle price enumeration. Additionally, there is a compelling question of designing mechanisms where we allow agents to select the best bundle efficiently, potentially in $poly(m)$ time, as suggested by previous research (Schapira & Singer, 2008). Our current research focused on a fixed order of agent visits, which prompts the exploration of methods to dynamically learn the optimal order in which agents should be visited (Brero et al., 2021). Extending our framework to accommodate non-deterministic allocation poses an intriguing challenge, and understanding how it can be implemented in the sequential setting needs further attention. Lastly, it is interesting to assess the ability of our approaches to approximate the non-sequential version of the auction problem. Innovations in this space could involve leveraging AMA-based approaches instead of RochetNet, which would engage with several agents in each step of a sequential auction instead of just one agent.

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

## A  DSIC AND IR

**Remark 1.** The mechanism described in Algorithm 1 is Dominant Strategy Incentive Compatible (IC) and Individual Rational (IR) if $\xi_{i,S}(\varnothing) = 0$ for all $i \in [n], S \subseteq M$.

An auction is *dominant strategy incentive compatible (DSIC)*, if each bidder's utility is maximized by reporting truthfully no matter what the other bidders report. An auction is *individually rational (IR)* if each bidder receives a non-negative utility while reporting truthfully.

The Sequential Combinatorial Auction (SCA) with menus is DSIC because the agents pick their utility maximizing bundle. Additionally, the utility of taking part is at least 0 (this is because the agent has the option to pick the empty bundle $\varnothing$ and is charged 0)

## B    PROOF OF PROPOSITION 1

**Proposition 2.** *For a current policy $\pi$ and value function $V_\pi(.)$, the improved policy $\pi'$ for a state $s^t$ is given by:*

$$\pi'(s^t) = \arg\max_{a^t} \mathbb{E}_{v \sim \mathcal{V}_{it}} \left[ a^t_{S^t_*(v)} + V_\pi(i^{t+1}, S^t \setminus S^t_*(v)) \right] \tag{1}$$

*Proof.* For a current policy $\pi$ and value function $V_\pi(.)$, the improved policy $\pi'$ for a state $s^t$ is given by:

$$\pi'(s^t) = \arg\max_{a^t} \sum_{s^{t+1}} p(s^{t+1}|s^t, a)[r(s^t, a, s^{t+1}) + V_\pi(s^{t+1})]$$

$$= \arg\max_{a^t} \sum_{S^{t+1} \subseteq S^t} \hat{p}(S^t \setminus S^{t+1}|s^t, a)[a^t_{S^t \setminus S^{t+1}} + V_\pi(s^{t+1})]$$

$$= \arg\max_{a^t} \sum_{T \subseteq S^t} \hat{p}(T|s^t, a)[a^t_T + V_\pi(i^{t+1}, S^t \setminus T)]$$

$$= \arg\max_{a^t} \mathbb{E}_{v \sim \mathcal{V}_{it}} \left[ \sum_{T \in S^t} \hat{p}(T|v, s^t, a) \left[ a^t_T + V_\pi(i^{t+1}, S^t \setminus T) \right] \right]$$

$$= \arg\max_{a^t} \mathbb{E}_{v \sim \mathcal{V}_{it}} \left[ a^t_{S^t_*(v)} + V_\pi(i^{t+1}, S^t \setminus S^t_*(v)) \right]$$

Here, $p(s^{t+1}|s^t, a)$ denotes the probability of the next state being $s_{t+1}$ when the current state is $s^t$ and $a$ being the action taken (i.e. prices). $\hat{p}(T|s^t, a)$ is the probability of bundle $T$ is picked at $s^t$ under pricing function $a$. When $v, s^t$ and, $a$ are known, we have $\hat{p}(T = S^t_*(v)|v, s^t, a) = 1$ where $S^t_*(v) = \arg\max_{T \subseteq S^t} v(T) - a^t_T$. $\qquad\square$

## C    IMPLEMENTATION DETAILS AND HYPERPARAMETERS

We use the stable-baselines3 (Raffin et al., 2021) package to implement our baselines.

**Actor Critic Networks.**    For all our neural networks, we use a simple fully connected neural network with *Tanh* activation functions except for the last layer. We use $R = 3$ hidden layers with $k = 256$ hidden units each.

For the actor network in PPO, we use *sigmoid* activation functions to squash the output to $[0, 1]$ range. The action distribution is a normal distribution with these sigmoid outputs as means. It is then scaled appropriately. For example, consider setting A with $n$ agents and $m$ items. The action space, which comprises of bundle prices, is of size $2^m$. The maximum possible valuation for a subset $T$ would simply be $|T|$. Consequently, we scale the output corresponding to the price of $T$ by $|T|$. We found this this approach performed better than using the *SquashedDiagonalGaussian* distribution where the noise is added before squashing the outputs using a *tanh* function. For the actor network in our approach, we only use a *softplus* activation to ensure the outputs are positive and leave them unscaled.

We make similar modifications for our large-scale settings involving entry-fee menu structure. For these settings, we use *sigmoid* functions for the posted price and a *softplus* function for the entry-fee for the actor networks in PPO as well as our approach.

We also found the using an offset before using *sigmoid* or *softplus* functions helped with training. This ensures that the prices start low but increase gradually. We offset sigmoid function by $0.5$ and softplus function by $1$. Thus, these modified functions are given by:

$$\textit{sigmoid}\text{-with-offset} = \frac{1}{1 + e^{-(x-0.5)}} \tag{4}$$

$$\textit{softplus}\text{-with-offset} = \log(1 + e^{(x-1)}) \tag{5}$$

| Hyperparameter | Value |
|:---:|:---:|
| $\ell$ | $2^{15}$ |
| $\tau$ | 100 |
| $\eta$ | 0.001 |
| $\Gamma$ | 2000 |

**Table 4:** Hyperparameters for Dynamic Programming (DP)

| Hyperparameter | Value |
|:---:|:---:|
| `num_iterations` | 20 |
| `num_environments` | 1024 |
| `num_critic_steps` | $100 + 500$ |
| `num_actor_steps` | 50 |
| $\gamma$ | 1 |
| GAE-$\lambda$ | 0.95 |
| $\epsilon_0$ | $e^{-2}$ |
| $\ell$ | 256 |
| $\eta_v$ | 0.0001 |
| $\eta_\pi$ | 0.0001 |
| $\eta_\epsilon$ | $e^{-\frac{1}{4}}$ |
| $\tau$ | 100 |

**Table 5:** Hyperparameters for Fitted Policy Iteration (FPI). Targets were updated using Equation 2 after 100 critic steps. This was followed by another 500 critic steps. For setting F, we set the $\eta_\pi$ to 0.001.

**Training and Evaluation Sets** Since we have access to the value distributions, we sample valuations online while training. But for testing, we report our results on a fixed batch of 10000 profiles.

**Hyperparameters** We present the hyperparameters used in Dynamic Programming (DP) in Algorithm 2 and Fitted Policy Iteration (FPI) below:

## D TRAINING CURVES

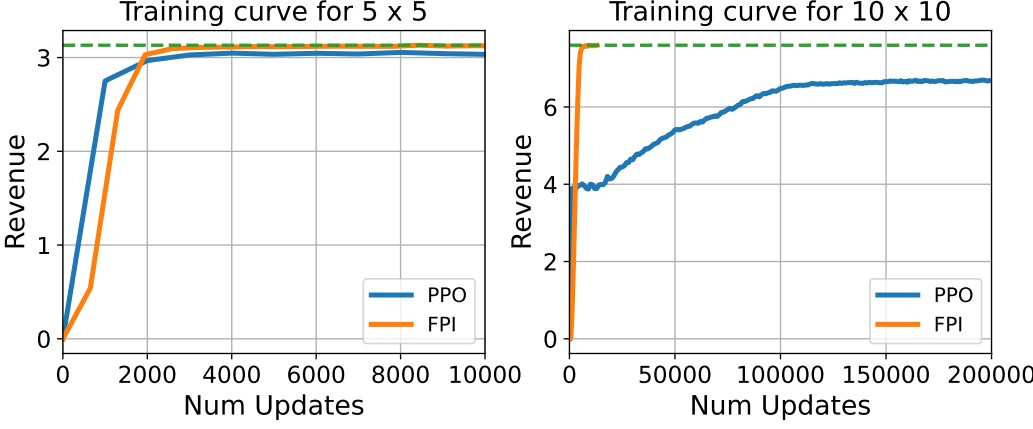

**Figure 2:** Training curves (expected revenue vs num updates) for Setting A: 5 agents and 5 items and 10 agents and 10 items.

The training curves for our approach (FPI) and PPO are presented in Figure 2. The plots illustrate the expected revenue (over 10,000 profiles) against the number of gradient updates applied, to the policy or value network (simultaneously in the case of PPO).

For Setting A, with 5 agents and 5 items, PPO attains a revenue of 3.09, whereas our approach achieves a slightly higher revenue of 3.12. For the same setting with 10 agents and 10 items, PPO

achieves a revenue of $6.68$ (we terminated PPO after 200000 updates which took about 5 hours). Our approach achieves a revenue of $7.59$ in less than 20 minutes.

