# OpenReview forum: "Deep Reinforcement Learning for Sequential Combinatorial Auctions"
_ICLR.cc/2025/Conference — Submitted to ICLR 2025_

### Official Review · Reviewer_zdTA · 2024-11-01

**Soundness:** 3
**Presentation:** 3
**Contribution:** 3
**Rating:** 6
**Confidence:** 3

**Summary:**

This paper introduces a new approach that uses fitted policy iteration and analytical gradients for learning revenue-maximizing sequential combinatorial auctions. Using twofold manner: initially refining the value function to align with the policy function and subsequently adjusting the policy function to maximize rewards. This method also can used in continuous action spaces via gradient descent.

**Strengths:**

This paper introduces a new approach that uses fitted policy iteration and analytical gradients for learning revenue-maximizing sequential combinatorial auctions. Using twofold manner: initially refining the value function to align with the policy function and subsequently adjusting the policy function to maximize rewards. This method also can used in continuous action spaces via gradient descent.

**Weaknesses:**

No

**Questions:**

No

---

### Official Review · Reviewer_hp6c · 2024-11-03

**Soundness:** 3
**Presentation:** 3
**Contribution:** 3
**Rating:** 6
**Confidence:** 2

**Summary:**

The paper explores designing revenue-maximizing mechanisms in sequential combinatorial auctions (SCAs) using deep reinforcement learning (DRL). SCAs present auction items sequentially, allowing bidders to make strategic choices at each stage. Although traditional sequential auctions have theoretical guarantees, they lack mechanisms to maximize revenue in complex environments. This paper addresses these limitations by proposing a deep reinforcement learning framework that leverages analytical gradients to optimize auctions involving multiple agents and items efficiently.

**Strengths:**

1. The paper provides theoretical foundations for its policy optimization approach and demonstrates its effectiveness through extensive experiments.
2. The paper introduces a new way to handle DRL in SCAs, particularly in combinatorial and high-dimensional settings.

**Weaknesses:**

* The method involves fitted policy iterations and analytical gradients. The complexity can be increased and needs to be measured.
* The framework assumes knowledge of agents' valuation distributions, which may not always be accessible or accurate in practice.

**Questions:**

1. What is the complexity of the new proposed method?
2. How to reduce the reliance on agents' valuation distributions?

---

> ### Author Response · Authors · 2024-11-21
>
> Thank you for your thoughtful review and positive assessment of our work.
>
> **Complexity:**
>
> Can you elaborate on what you mean by complexity?
>
> In Appendix D, we include the training curves which plot the expected revenue with respect to the number of updates. Our approach converges much more quickly compared to PPO in terms of the number of updates as well as time taken. For the 10 agents and 10 items setting, our method required 20 minutes. We terminated PPO at around 5 hours.
>
> ---
>
> **Reliance on agents' valuation distributions:**
>
> Please refer to the comment [here](https://openreview.net/forum?id=SVd9Ffcdp8&noteId=rbvZgDpl8R).

---

> > ### Comment · Reviewer_hp6c · 2024-11-27
> >
> > I thank the authors for responding to reliance on agents' valuation distribution.
> >
> > By complexity, I mean the run time comparison to the mentioned baselines in the paper. I wonder if the new algorithm will greatly increase the run time compared to the baselines.

---

> > > ### Author Response · Authors · 2024-11-28
> > >
> > > Thank you for your question!
> > >
> > > We have detailed the runtimes in Section 4 under the respective settings, and here is a summary of the results:
> > >
> > > **FPI (Ours):** Completed training in under 20 minutes across all settings.
> > > **DP (Ours):** Required 6 hours for the constrained additive setting and 10 hours for the combinatorial setting, as it involves running RochetNets for every state.
> > > **PPO (RL Baseline):** Terminated after 5 hours for the constrained additive setting and 12 hours for the combinatorial setting.
> > >
> > > Overall, FPI outperforms PPO in both time and performance, while achieving results comparable to DP with a substantial reduction in runtime.

---

### Official Review · Reviewer_PQ2q · 2024-11-04

**Soundness:** 2
**Presentation:** 3
**Contribution:** 2
**Rating:** 6
**Confidence:** 2

**Summary:**

This paper proposed a method on designing revenue-optimal mechanisms for sequential combinatorial auctions to allocate items to multiple agents in stages where the agents place bids on bundles of items based on their preferences using DRL with designs like analytical gradients.

**Strengths:**

- The paper is generally well-written and easy to follow, the motivation on solving limitations from RL on sample inefficiency and issues with convergence in large, continuous action spaces is valid
- Use of Analytical Gradients seems to be effective in enhancing sample efficiency and convergence.
- The approach demonstrates scalability to scenarios with empirical results.

**Weaknesses:**

-  Despite improvements, the method may still face computational challenges in extremely large-scale auctions, similar to issues noted in Pieroth et al. (2023).
- The reliance on known valuation distributions may limit applicability in settings where such information is unavailable which limits the use cases of this method, more explanation on this would be good
- The baseline selection needs to be justified, why not also compare with more state-of-the-art algorithms?
- This approach uses a fixed order of agent visits, which can limit its optimality in situations where the order of bidding impacts the auction’s overall revenue.
- Whether and how much this work is sensitive to hyperparameters is not explained.

**Questions:**

Please see weakness.

---

> ### Author Response · Authors · 2024-11-13
> **Baselines**
>
> Thank you for your thoughtful review!
>
> We appreciate your suggestions and would love to know if there are specific baselines you have in mind. This way, we can potentially incorporate additional experiments and update our results accordingly. We’ll address the remaining points on weaknesses shortly.
>
> **Choice of Baselines:**
> We focused on PPO and SAC, as they are the state-of-the-art methods for continuous action spaces. As noted in Section 1, DQN and other Q-learning approaches are suitable only for discrete action spaces, which limits their applicability here. TRPO, while valuable, is considered a predecessor to PPO, and it tends to be less sample-efficient and slower in practice. Empirically, PPO often demonstrates superior performance, which is why we chose to prioritize it over TRPO in our results. We also tested DDPG in some settings, but it proved unstable in our experiments.

---

> ### Author Response · Authors · 2024-11-21
>
> Thank you for your detailed review and thoughtful feedback!
>
> **Scaling:**
> Our settings are quite different from Pieroth et. al. (they focus on computing equilibrium strategies). Our focus is on the inverse problem (given truth telling is the equilibrium, what are the rules of a revenue-optimal auction).
>
> Our approach demonstrates strong scalability, achieving results for settings with up to 50 agents and 50 items—far exceeding the capabilities of existing differentiable economics-based methods for auction design.
>
> _Scaling Further_: The primary limitation of running RochetNet with all possible bundle prices arises from hardware constraints. For m > 10 items, the need to output 2^m bundle prices introduces significant memory and computational demands. However, if we restrict ourselves to subadditive valuations, we can scale up to m = 50 items, with scaling linear in terms of space and O(m log m) in terms of computation.
>
> We can, however, expand scalability in terms of the number of bidders while keeping the number of items fixed. For example, with 200 bidders, we can process them in batches of 50—starting with the last 50 bidders, followed by the preceding 50, and so on. This strategy serves as a natural hybrid between our DP and FPI approaches, allowing us to achieve optimal results. Importantly, this method is constrained only by computational time and not memory.
>
> ---
>
> **Reliance on value distribution:**
> Please refer to the comment [here](https://openreview.net/forum?id=SVd9Ffcdp8&noteId=rbvZgDpl8R)
>
> ---
>
> **Baselines:**
>
> We would love to know if there are specific baselines you have in mind.
>
> We focused on PPO and SAC, as they are the state-of-the-art methods for continuous action spaces. As noted in Section 1, DQN and other Q-learning approaches are suitable only for discrete action spaces, which limits their applicability here. TRPO, while valuable, is considered a predecessor to PPO, and it tends to be less sample-efficient and slower in practice. Empirically, PPO often demonstrates superior performance, which is why we chose to prioritize it over TRPO in our results. We also tested DDPG in some settings, but it proved unstable in our experiments.
>
> ---
>
> **Order of Agents:**
> If the bidder types (value distributions) are identical, the order doesn't affect the outcome as we are concerned with expected revenue. However, if they are not identical, we can train RochetNets at each state over an equal mixture of these types. This would add the benefit of being order agnostic!
>
> It is important to note that our focus on strategy-proof mechanisms prevents the use of agents' reports or dynamically realized orders to modify the menu, as doing so would compromise incentive compatibility.
>
>
> ---
> **Hyperparameters:**
> As shown in Table 5 in the Appendix, most hyperparameters were set to default values for all the settings.
> Empirical evidence from our experiments indicates that our approach is relatively robust to variations in hyperparameters. This robustness stems from its reliance on first-order gradients for updates, in contrast to traditional RL methods, which rely heavily on exploratory noise and involve tuning numerous hyperparameters.

---

> ### Comment · Reviewer_PQ2q · 2024-11-26
>
> I thank the authors for their response and increased my score. However, I still believe more domain-specific SOTA works should be included for baseline comparison

---

> ### Author Response · Authors · 2024-11-28
>
> Thank you for updating the score and engaging with us. If there are any domain-specific state-of-the-art methods you believe we should consider, we would greatly appreciate your suggestions.
>
> Much of the existing work in this domain consists of existential results, with constructive approaches typically limited to highly restrictive settings. In this paper, we have introduced what we believe to be the first general-purpose approach to designing mechanisms for the sequential auction setting. In addition to comparing against standard mechanism baselines (e.g., selling item-wise and bundle-wise), we have also evaluated our method against standard reinforcement learning approaches such as PPO and SAC. We would be happy to incorporate further baselines if you have any additional recommendations.

---

### Official Review · Reviewer_W499 · 2024-11-12

**Soundness:** 3
**Presentation:** 3
**Contribution:** 2
**Rating:** 5
**Confidence:** 4

**Summary:**

This work investigates sequential combinatorial auctions, an instance of auction where the bidders arrive in a sequence and place bids on bundles (combinations) of items. Sequential combinatorial auctions is relevant due to its simplicity, its strategyproofness, and its generality. This work aims on numerically finding the optimal solution. Previously, this was achieved by formulating the problem into an MDP, and then running PPO on it. But the previous work was under the additive assumption so it did not really tackle the combinatorial nature of the problem. This obviously skipped the exponential action space brought by the bundles.

This work leverages the particular structure of auction, and proposes a gradient that uses the knowledge of the world model. This allows a first-order gradient feedback of the update. The work is inspired by fitted policy iteration (which I'm not sure why the manuscript did not cite any previous work) and the policy improvement step of which could benefit from the gradient derived.

With some additional techniques and tricks, the work is capable of hosting a decent size (at most 50 items) problem. Experiments are conducted on both additive and combinatorial settings against baselines including mechanism design methods and the previous RL approach. The proposed method seems to work well on the experiments.

**Strengths:**

1. This work numerically solves sequential auctions with combinatorial action space. The solution is through an organic synergy between policy gradient and the auction process.
2. It scales to an action space with as many as 50 items.
3. Experiments show that it outperforms previous baselines.

**Weaknesses:**

1. The topic (sequential auction + combinatorial + numerical solution) is a bit limited to the specific sub-community. I'm not seeing the method/techniques to be of a general interest.
2. The experiments are conducted only on toy examples. Given the numerical nature of the work, I was expecting some real data, or even real system, experiments.

**Questions:**

Isn't FTI an existing algorithm? I don't see a citation on that.

---

> ### Author Response · Authors · 2024-11-21
>
> Thank you for your detailed review and thoughtful feedback!
>
> **On Relevance and General Interest**
> Our work contributes to the emerging field of AI for Economics Theory, showcasing how reinforcement learning can effectively solve complex mechanism design problems that are analytically intractable.
>
> There is an existing body of literature regarding the use of deep learning for the design of multi-item auctions. This area has garnered significant interest within the machine learning community, as evidenced by numerous papers presented at conferences such as ICML, NeurIPS, and ICLR. Notably, works like "Optimal Auctions through Deep Learning" at ICML-19 and others such as [Curry et al., NeurIPS-20], [Rahme et al., ICLR-21], [Duan et al., ICML-22], [Ivanov et al., NeurIPS-22], and [Duan et al., NeurIPS-23] have contributed to the exploration of deep learning approaches for improving multi-item auction design, as cited in our paper.
>
> In addition, we want to highlight that our approach shows promise for _broader applicability_. For problems featuring large action and continuous spaces, wherever we can model transitions differentiably (for instance in physics or robotics), we anticipate our approach outperforming existing standard RL methods. In this paper, we employ multiple RochetNets to compute the policy (which is most suited for learning menu-based mechanisms). However, for a general MDP, alternative methods could be explored. For instance, one could use linear programming (LP) or convex optimization to compute optimal policies in the continuous space, leverage gradient computation techniques proposed by [1] to obtain gradients through the solutions, and integrate this with fitted policy iteration.
>
> **References**
> [1] Agrawal, A., Amos, B., Barratt, S., Boyd, S., Diamond, S., & Kolter, J. Z. (NeurIPS 2019).
> Differentiable convex optimization layers
>
> ---
>
> **Real-world examples**
> Sequential auctions are frequently used in real life for their simplicity and strategy-proofness guarantees. For instance, companies like Costco and Amazon use Sequential Mechanisms with Entry Fees. To the best of our knowledge, we didn’t find any publicly available dataset (as most of these datasets are proprietary). If you know any please suggest them and we will include them. As a fallback, we have endeavored to capture some real-world structures through different distributions (symmetric, asymmetric) and different valuation types (additive, k-demand, combinatorial).
>
> ---
>
> **FPI citation**
> Thanks for pointing this out. We have updated our paper with the required citations for FPI.
>
> ---

---

> ### Comment · Reviewer_W499 · 2024-11-24
>
> Thank you for providing a rebuttal. I agree that AI for Economics Theory is an emerging topic. It is the specific combination of three (sequential auction + combinatorial + numerical solution) that makes the topic niche. As you discussed companies like Costco and Amazon use Sequential Mechanisms with Entry Fees, could you provide enough evidence that this is exactly how they work and it fits close enough to your setting/your combination of three?

---

> > ### Author Response · Authors · 2024-11-28
> >
> > Sequential mechanisms with entry fees are a simplification of real-world practices, but they capture some key aspects of how Costco and Amazon operate.
> > - Costco [1] and Amazon Prime [2] both charge annual and monthly membership fees respectively, which acts as an _entry fee_ to access their offerings.
> > - Costco is known for selling items in large quantities. Both the companies offer discounts on bundles. This captures _Combinatorial Preferences_.
> > - Limited-time deals (called Lightning Deals on Amazon [3]) and Dynamic pricing [4] make it a _Sequential model_ as well.
> >
> > While publicly available details about proprietary mechanisms are limited, this framework aligns with sequentially offering bundles or individual items to maximize revenue. Our paper identifies the challenges in addressing such problems using standard reinforcement learning techniques and proposes a novel approach to overcome these limitations!
> >
> >
> > **References**
> > [1] https://www.costco.com/join-costco.html
> >
> > [2] https://www.amazon.com/gp/help/customer/display.html?nodeId=G34EUPKVMYFW8N2U
> >
> > [3] https://www.amazon.com/gp/help/customer/display.html?nodeId=GW3L8JX7Q9FH8ALB
> >
> > [4] Price tracker for Amazon: https://camelcamelcamel.com/

---

### Author Response · Authors · 2024-11-21
**Reliance on Samples from Valuation Distribution**

The reliance on samples from valuation distributions is a standard assumption in the automated mechanism design literature, both in deep learning-based approaches [1] and traditional methods [2]. In the context of independent private value auctions, as considered in this paper, the auctioneer only has access to valuation distributions or samples derived from them, with the objective of designing a mechanism that maximizes expected revenue while maintaining incentive compatibility and individual rationality. For a comprehensive discussion on the sample complexity required to ensure, with high probability, that the average revenue over the sample approximates the expected revenue with respect to the underlying unknown distribution, we refer the reviewers to [3]. However, it is worth noting that these results are specific to AMA auctions and do not readily extend to deep learning methods.

To address this, we aim to empirically evaluate the reliance on samples through an additional experiment. Specifically, we train both PPO and our approach using a fixed number of samples (s = {2, 5, 10, ..., 10000}) drawn from the valuation distribution, instead of sampling online, and report their performance on the same test set used originally. This provides an empirical perspective on the sample complexity requirements of our method and PPO. The results are in Tables 1 and 2.

| **Num Samples** | **PPO (rev)** | **FPI (rev)** |
|:--------------------:|:-------:|:-------:|
|          2           |   2.43  |   2.01  |
|          5           |   2.60  |   2.68  |
|         10           |   2.88  |   2.66  |
|        100           |   3.04  |   3.06  |
|       1000           |   3.08  |   3.12  |
|      10000           |   3.08  |   3.12  |
| online                | 3.09 | 3.12 |

**Table 1:**  Performance with varying number of samples for Setting A with 5 agents and 5 items

| **Num Samples** | **PPO (rev)** | **FPI (rev)** |
|:--------------------:|:-------:|:-------:|
|          2           |  5.85  |   5.33  |
|          5           |   5.45  |   6.25  |
|         10           |   5.37  |   6.80  |
|        100           |   6.51  |   7.42  |
|       1000           |   6.61  |   7.56  |
|      10000           |   6.68  |   7.58  |
| online                | 6.68 | 7.59 |

**Table 2:**  Performance with varying number of samples for Setting A with 10 agents and 10 items

---

**Reference**
[1] Dütting, P., Feng, Z., Narasimhan, H., Parkes, D. C., & Ravindranath, S. S. (2024). Optimal auctions through deep learning: Advances in differentiable economics. Journal of the ACM, 71(1), 1-53.

[2] Sandholm, Tuomas. "Automated mechanism design: A new application area for search algorithms." International Conference on Principles and Practice of Constraint Programming. Berlin, Heidelberg: Springer Berlin Heidelberg, 2003.

[3] Balcan, M. F. F., Sandholm, T., & Vitercik, E. (NeurIPS 2016).
Sample complexity of automated mechanism design

---

### Meta-Review · Area_Chair_e4Ho · 2024-12-21

**Metareview:**

This paper applies deep reinforcement learning (DRL) to the problem of revenue-maximizing sequential combinatorial auctions (SCAs), leveraging analytical gradients and fitted policy iteration to address sample efficiency and scalability. However, the approach appears to be a direct application of standard DRL techniques tailored to the auction setting by exploiting specific transition structures, raising concerns about its broader contribution to DRL literature and real-world applicability. While the authors addressed scalability and baseline selection, the relevance to real-world systems remains unclear, and the work does not convincingly demonstrate novel contributions to the DRL community beyond applying existing methods to a specific problem.

**Additional Comments On Reviewer Discussion:**

During the rebuttal period, Reviewer W499 raised concerns about scalability and relevance to real-world systems, which the authors addressed by discussing batching techniques and aligning their approach with practical mechanisms like those used by Amazon and Costco. Reviewer PQ2q highlighted the need for additional domain-specific baselines, and while the authors justified their baseline choices but did not include any domain-specific benchmark comparisons. Reviewer hp6c questioned runtime complexity, which the authors clarified with detailed comparisons, and reliance on valuation distributions, which they defended as reasonable assumptions. I echo with the concerns raised by Reviewer W499 and moreover, I have some concerns about the novelty of the overall approach.

---

### Decision · Program_Chairs · 2025-01-22

Reject